# Diagnostic performance of the GenoType MTBDR*plus* VER 2.0 line probe assay for the detection of isoniazid resistant *Mycobacterium tuberculosis* in Ethiopia

Shewki Moga[1,2]*, Kidist Bobosha[3], Dinka Fikadu[1], Betselot Zerihun[1], Getu Diriba[1], Misikir Amare[1], Russell R. Kempker[4], Henry M. Blumberg[4,5], Tamrat Abebe[2]

1 Ethiopian Public Health Institute (EPHI), Addis Ababa, Ethiopia, 2 Department of Microbiology, Immunology, and Parasitology, School of Medicine, College of Health Sciences, Addis Ababa University (AAU), Addis Ababa, Ethiopia, 3 Armauer Hansen Research Institute (AHRI), Addis Ababa, Ethiopia, 4 Department of Medicine, Division of Infectious Diseases, Emory University School of Medicine, Atlanta, Georgia, United States of America, 5 Departments of Epidemiology and Global Health, Emory Rollins School of Public Health, Atlanta, Georgia, United States of America

* shewki2002@gmail.com

**Data Availability Statement:** All relevant data are within the paper and its Supporting information files.

## Abstract

### Background

Isoniazid (INH) resistant *Mycobacterium tuberculosis* (Hr-TB) is the most common type of drug resistant TB, and is defined as *M tuberculosis* complex (MTBC) strains resistant to INH but susceptible to rifampicin (RIF). Resistance to INH precedes RIF resistance in almost all multidrug resistant TB (MDR-TB) cases, across all MTBC lineages and in all settings. Therefore, early detection of Hr-TB is critical to ensure rapid initiation of appropriate treatment, and to prevent progression to MDR-TB. We assessed the performance of the GenoType MTBDR*plus* VER 2.0 line probe assay (LPA) in detecting isoniazid resistance among MTBC clinical isolates.

### Methods

A retrospective study was conducted among *M. tuberculosis* complex (MTBC) clinical isolates obtained from the third-round Ethiopian national drug resistance survey (DRS) conducted between August 2017 and December 2019. The sensitivity, specificity, positive predictive value, and negative predictive value of the GenoType MTBDR*plus* VER 2.0 LPA in detecting INH resistance were assessed and compared to phenotypic drug susceptibility testing (DST) using the Mycobacteria Growth Indicator Tube (MGIT) system. Fisher's exact test was performed to compare the performance of LPA between Hr-TB and MDR-TB isolates.

### Results

A total of 137 MTBC isolates were included, of those 62 were Hr-TB, 35 were MDR-TB and 40 were INH susceptible. The sensitivity of the GenoType MTBDR*plus* VER 2.0 for

**Funding:** This study was supported in part by a grant from the U.S. National Institutes of Health (NIH) Fogarty International Center (D43TW009127). The funder had no role in study design, data collection and interpretation, or the decision to submit the work for publication.

**Competing interests:** The authors have declared that no competing interests exist.

detecting INH resistance was 77.4% (95% CI: 65.5–86.2) among Hr-TB isolates and 94.3% (95% CI: 80.4–99.4) among MDR-TB isolates ($P$ = 0.04). The specificity of the GenoType MTBDR*plus* VER 2.0 for detecting INH resistance was 100% (95% CI: 89.6–100). The *katG* 315 mutation was observed in 71% (n = 44) of Hr-TB phenotypes and 94.3% (n = 33) of MDR-TB phenotypes. Mutation at position-15 of the *inh*A promoter region alone was detected in four (6.5%) Hr-TB isolates, and concomitantly with *katG* 315 mutation in one (2.9%) MDR-TB isolate.

## Conclusions

GenoType MTBDR*plus* VER 2.0 LPA demonstrated improved performance in detecting INH resistance among MDR-TB cases compared to Hr-TB cases. The *kat*G315 mutation is the most common INH resistance conferring gene among Hr-TB and MDR-TB isolates. Additional INH resistance conferring mutations should be evaluated to improve the sensitivity of the GenoType MTBDR*plus* VER 2.0 for the detection of INH resistance among Hr-TB cases.

## Introduction

A major challenge in global tuberculosis (TB) control is the prevention and management of drug-resistant TB (DR-TB) [1]. While multidrug-resistant TB (MDR-TB), defined as resistance to isoniazid (INH) and rifampicin (RIF), has garnered the most global attention, less has been provided to INH-resistant TB (Hr-TB). Hr-TB is the most common type of drug resistant TB and is defined as *M tuberculosis* complex (MTBC) strains resistant to INH but susceptible to RIF [2]. In 2019, Hr-TB occurred in an estimated 1.1 million (11%, range 6.5–15%) people with TB, which is a much higher annual burden than the estimated 465,000 RIF-resistant TB (RR-TB) or MDR-TB cases [1]. Hr-TB cases have higher chances of amplification to MDR-TB, treatment failure, and TB relapse rates compared to drug susceptible TB [3]. Moreover, resistance to INH precedes RIF resistance in almost all MDR-TB cases, across all MTBC lineages, and in all settings [4]. Therefore, early detection and management of Hr-TB is critical to ensure rapid initiation of appropriate treatment, helping to prevent treatment failure, the emergence of additional drug resistance including progression to MDR-TB.

Phenotypic drug susceptibility testing (DST) is considered the gold standard for detection of drug resistant MTBC. However, it must be performed on a pure culture of MTBC isolates and requires up to 4–6 weeks for results [5]. Phenotypic DST also requires complex and expensive infrastructure, and highly trained and skilled personnel. Major progress has been made in deciphering the genotypic basis of MTBC, which has resulted in the development of rapid molecular tests [6]. Some of these technologies have the added advantage of being able to detect resistance to selected anti-TB drugs with high sensitivity and specificity [7]. The WHO has approved several such molecular diagnostic tests, including Xpert MTB/RIF assay and Truenat MTB-RIF Dx to detect RIF resistance, and Genotype MTBDR*plus* line probe assay (LPA) to detect resistance to RIF and INH [8].

The line probe assay (LPA), known as GenoType MTBDR*plus* (Hain Life science GmbH, Nehren, Germany), is a rapid molecular diagnostic test that is used for the detection of MTBC and resistance to INH and RIF [9]. The first version, Genotype MTBDR*plus* v1, was validated for smear-positive sputum and culture positive isolates only [7]. However, the second version,

Genotype MTBDR*plus* VER 2.0, can additionally be performed on smear-negative clinical specimens given its higher sensitivity than version-1 [10]. The assay identifies INH resistance by detecting mutations in the *katG* and *inh*A genes, which are thought to be responsible for most phenotypic resistance to INH. However, a growing body of literature has demonstrated that the distribution of these genes varies across countries, and between Hr-TB and MDR-TB isolates [11–13]. Given this variability and the limited data from certain geographic regions, we sought to evaluate the diagnostic performance of LPA in detecting INH resistance among Hr-TB cases for determining its implementation in Ethiopia. We also sought to investigate variants and frequency of *kat*G and *inh*A mutations among Hr-TB cases using LPA.

## Methods

The retrospective study was conducted among *M. tuberculosis* complex (MTBC) clinical isolates obtained from the third Ethiopian national drug resistance survey (DRS) conducted between August 2017 and December 2019. The DRS is a nationally representative multi-center survey conducted every five years to assess the drug resistance pattern of MTBC isolates among new and previously treated pulmonary TB cases in Ethiopia. For isolation of MTBC, smear positive or Xpert MTB/RIF positive sputum specimens were digested and decontaminated with N-acetyl-L-cysteine-sodium hydroxide method as previously described (13). The processed specimen were inoculated into Löwenstein-Jensen (LJ) solid culture media and Mycobacteria Growth Indicator Tube (MGIT) liquid culture media for cultivating mycobacteria. Positive cultures were subjected to SD Bioline TB Ag MPT4 rapid testing (Standard Diagnostics, Yongin, South Korea) for identification of MTBC isolates.

Both LPA and phenotypic DST were conducted on pure isolates of MTBC at the National TB Reference Laboratory (NTRL), which is part of the Ethiopian Public Health Institute (EPHI) in Addis Ababa. Phenotypic DST was performed using the BACTEC MGIT 960 (Becton Dickinson, Sparks, USA) according to the WHO-recommended critical concentrations of 0.1 μg/ml for INH, 1.0 μg/ml for RIF, 1.0 μg/ml for streptomycin (STR), 5 μg/ml for ethambutol (EMB), and 100 μg/ml for pyrazinamide (PZA) [14].

GenoType MTBDR*plus* LPA VER 2.0 (Hain LifeScience GmbH, Nehren, Germany) was performed according to the manufacturer's instruction. Briefly, DNA was extracted from MTBC culture isolates using a GenoLyse extraction kit. The extract was then amplified with a polymerase chain reaction (PCR). Following amplification, amplicons were hybridized with specific oligonucleotide probes immobilized on a strip resulting in a colored band pattern. INH resistance was detected using wild type (WT) probes and common mutation (MUT) probes for *kat*G 315 and *inh*A genes. Isolates were considered INH resistant if color bands for *kat*G MUT and *inh*A MUT probes were detected, or if color bands for the *kat*G WT and *inh*A WT probes were missing. Isolates were considered susceptible if all WT probes were detected and MUT probes were not detected. The simultaneous hybridization of all WT probes and one of the MUT probe in any gene indicates the presence of hetero-resistance and the result were interpreted as resistant. The specific nucleotide changes for each detected resistance mutation was also interpreted [15].

Laboratory staff performing the LPA were blinded to the results of the phenotypic DST and patient clinical data. Similarly, staff performing phenotypic DST were blinded to LPA results and patient clinical data. *M. tuberculosis* strain H37R$_V$ was used as a positive control for LPA and phenotypic DST. DNA-free molecular grade water and blank reagent were used as negative controls for LPA. The EPHI/NTRL participates in the External Quality Assessment (EQA) scheme provided by the Supranational Reference Laboratory (SRL, Italy and Uganda). In the 2021 EQA performance record, the NTRL demonstrated 100% agreement in LPA testing and

98.3% agreement in phenotypic DST proficiency for both first-line drugs and second-line drugs.

## Sample size

The sample size for assessing the diagnostic performance of LPA in detecting INH resistance was determined as described by Banoo *et al.* using the $n \geq \frac{(1.96)^2 p(1-p)}{x^2}$ formula; where $p$ = sensitivity or specificity, and $x$ = desired precision with which the sensitivity or specificity to be measured, [16]. By using a sensitivity of 89.4% for detecting INH resistance [17] and precision of ± 10%, a total of 36 phenotypically INH resistant Hr-TB isolates were required for assessing the sensitivity of GenoType MTBDR*plus* LPA VER 2.0. Forty INH susceptible phenotypes were randomly selected and included in the study to assess the specificity of LPA. MDR-TB phenotypes were also included to compare the performance of LPA and frequency of *kat*G and *inh*A mutations between MDR-TB and Hr-TB isolates.

## Data analysis

The diagnostic performance of LPA for detecting INH resistance was evaluated using phenotypic DST results as the gold standard. Resistance data were analyzed using SPSS version 20 (IBM Corp., Armonk, N.Y., USA) and online MedCalc tool (https://www.medcalc.org/calc/diagnostic_test.php). The diagnostic performance of the LPA was assessed by performing sensitivity, specificity, positive predictive value, and negative predictive value, and calculating 95% confidence intervals (95% CIs). Fisher's exact test was performed to compare the sensitivity of LPA between Hr-TB and MDR-TB and difference with $P$ value of <0.05 was considered statistically significant.

## Ethical consideration

The study protocol was approved by the Institutional Review Boards (IRBs) of the Ethiopian Public Health Institute and Addis Ababa University (AAU). Since the study involved secondary analysis of stored isolates, a waiver of individual informed consent was obtained from the IRBs. Informed consent was collected from participants enrolled in the primary DRS.

## Results

Phenotypic DST results of MTBC isolates were available for 68.4% (1552/2268) of DRS participants, of those Hr-TB and MDR-TB was identified in 4.0% (62/1552) and 2.3% (35/1552) of tested isolates, respectively. INH resistance was detected in 94.6% (35/37) of RIF resistant isolates (Fig 1). Overall, 6.3% (97/1552) INH resistant phenotypes were identified.

### Diagnostic performance of GenoType MTBDR*plus* VER 2.0 for the detection of INH resistance among INH resistant phenotypes

A total of 137 isolates, including 40 INH susceptible phenotypes, were used to evaluate the diagnostic performance of GenoType MTBDR*plus* VER 2.0 in detecting INH resistance both in MDR-TB and Hr-TB isolates (n = 97). The sensitivity of the assay for detection of INH resistance in MDR and Hr-TB isolates combined was 83.5% (95% CI: 74.8–89.7) (Table 1). The specificity of the assay was 100% (95% CI: 89.6–100). INH resistance mutations were not detected in any of the 40 INH susceptible MTBC isolates.

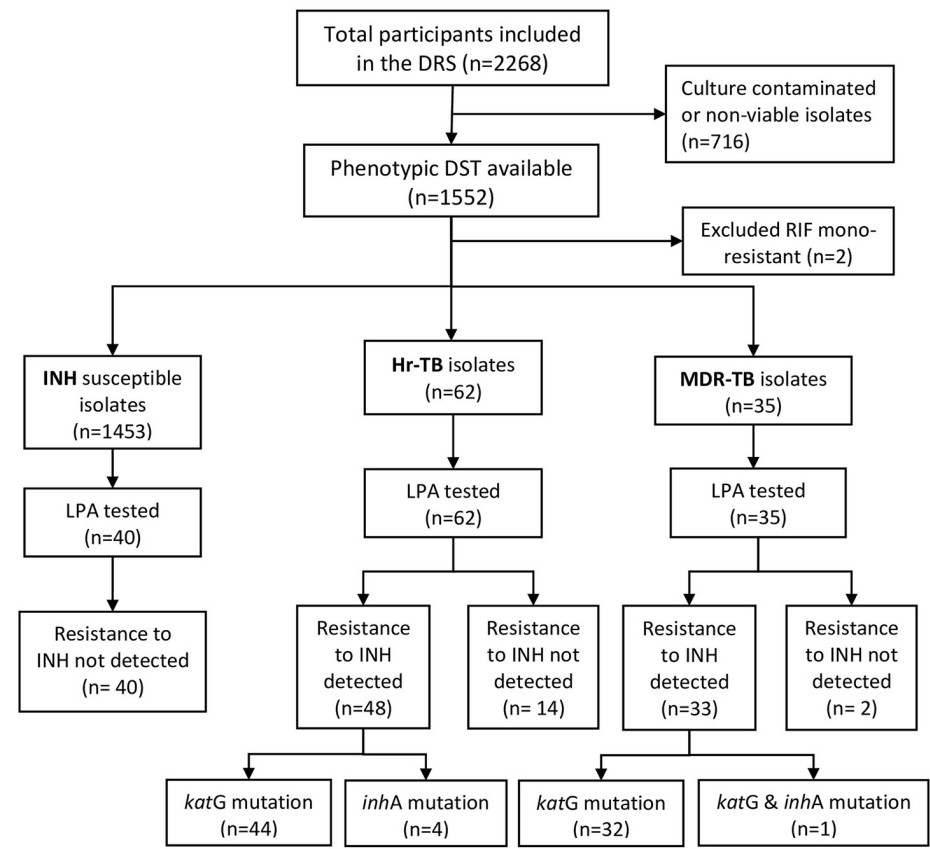

**Fig 1. Flowchart of drug resistance survey (DRS) study isolates used for diagnostic evaluation of LPA.** DST = drug susceptibility testing; INH = isoniazid; RIF = rifampicin; Hr-TB = Isoniazid resistant TB; MDR-TB = multi-drug resistant TB; LPA = line probe assay.

## Diagnostic performance of GenoType MTBDR*plus* VER 2.0 for the detection of INH resistance among Hr-TB cases

A total of 102 isolates were used to assess the diagnostic performance of GenoType MTBDR*plus* VER 2.0 for the detection of INH resistance among Hr-TB. The assay detected only 48 of the 62 Hr-TB identified by phenotypic DST resulting in a sensitivity of 77.4% (95% CI: 65.5–86.2). The sensitivity of LPA in detecting INH resistance among Hr-TB cases was significantly less than that of detecting INH resistance among MDR-TB cases (77.4% vs 94.3%, *P* = 0.04). The specificity of the assay was 100% (95% CI: 89.6–100) (Table 2).

**Table 1. Diagnostic performance of GenoType MTBDR*plus* VER 2.0 LPA for detecting INH resistance in culture of INH resistant phenotypes (Hr-TB and MDR-TB).**

| GenoType MTBDR*plus* VER 2.0 line probe assay | Phenotypic DST | | Total | Sensitivity (95% CI) | Specificity (95% CI) | PPV | NPV (95% CI) |
|---|---|---|---|---|---|---|---|
| | INH resistant | INH sensitive | | | | | |
| INH resistant | 81 | 0 | 81 | 83.5% (74.8–89.7) | 100% (89.6–100) | 100% | 98.9% (98.3–99.3) |
| INH sensitive | 16 | 40 | 56 | | | | |
| **Total** | **97** | **40** | **137** | | | | |

95% CI = 95% confidence interval; INH = isoniazid; PPV = positive predictive value; NPV = negative predictive value

**Table 2. Diagnostic performance of GenoType MTBDR*plus* VER 2.0 LPA for detecting isoniazid resistance in culture of Hr-TB isolates.**

| GenoType MTBDR*plus* VER 2.0 line probe assay | Phenotypic DST | | Total | Sensitivity (95% CI) | Specificity (95% CI) | PPV | NPV (95% CI) |
|---|---|---|---|---|---|---|---|
| | INH resistant | INH sensitive | | | | | |
| INH resistant | 48 | 0 | 48 | 77.4% (65.5–86.2) | 100% (89.6–100) | 100% | 99.0% (98.5–99.4) |
| INH sensitive | 14 | 40 | 54 | | | | |
| **Total** | **62** | **40** | **102** | | | | |

95% CI = 95% confidence interval; INH = isoniazid; PPV = positive predictive value; NPV = negative predictive value

**Table 3. Diagnostic performance of GenoType MTBDR*plus* VER 2.0 LPA for detecting isoniazid resistance in culture of MDR-TB isolates.**

| GenoType MTBDR*plus* VER 2.0 line probe assay | Phenotypic DST | | Total | Sensitivity (95% CI) | Specificity (95% CI) | PPV | NPV (95% CI) |
|---|---|---|---|---|---|---|---|
| | INH resistant | INH sensitive | | | | | |
| INH resistant | 33 | 0 | 33 | 94.3% (80.4–99.4) | 100% (89.6–100) | 100% | 50.0% (20.6–79.3) |
| INH sensitive | 2 | 40 | 42 | | | | |
| **Total** | **35** | **40** | **75** | | | | |

95% CI = 95% confidence interval; INH = isoniazid; PPV = positive predictive value; NPV = negative predictive value

## Diagnostic performance of GenoType MTBDR*plus* VER 2.0 for the detection of INH resistance among MDR-TB cases

A total of 75 isolates were used to assess the diagnostic performance of GenoType MTBDR*plus* VER 2.0 for detecting INH resistance among MDR-TB isolates. As compared to phenotypic DST, the sensitivity of the assay for the detection of INH resistance was 94.3% (95% CI: 80.4–99.4) (Table 3). The sensitivity of LPA in detecting INH resistance among MDR-TB cases was significantly greater than the sensitivity of LPA in detecting INH resistance among Hr-TB cases (94.3% vs 77.4%, $P$ = 0.4). The specificity of the assay was 100% (95% CI: 89.6–100).

## Resistance pattern of Hr-TB isolates to first line anti-TB drugs

All the 62 Hr-TB isolates were tested for phenotypic DST against streptomycin (STR), ethambutol (EMB) and pyrazinamide (PZA). Table 4 shows the phenotypic DST pattern of Hr-TB isolates identified from DRS to all first line anti-TB drugs. Thirty one Hr-TB isolates had additional resistance to first line anti-TB drugs and 31 Hr-TB isolates demonstrated mono resistance to INH. Among the Hr-TB isolates, additional resistance was observed to STR in 28

**Table 4. Susceptibility pattern of Hr-TB isolates to first line anti-TB drugs (n = 62).**

| INH resistance type | Pattern | Number (%) |
|---|---|---|
| INH mono-resistant TB | INH alone | 31 (50) |
| INH poly-resistant TB | INH-STR only | 21 (34) |
| | INH-PZA only | 2 (3.2) |
| | INH-EMB only | 1 (1.6) |
| | INH-STR-PZA only | 3 (4.8) |
| | INH-STR-EMB only | 2 (3.2) |
| | INH-STR-EMB-PZA | 2 (3.2) |

INH = isoniazid; STR = streptomycin; PZA = pyrazinamide; EMB = ethambutol

**Table 5. Frequency and variants of isoniazid resistance conferring mutations among Hr-TB and MDR-TB isolates detected using the GenoType MTBDR*plus* VER 2.0 assay.**

| Gene(s) | Mutation probe(s) | Nucleotide change | Hr-TB (n = 48) Number (%) | MDR-TB (n = 33) Number (%) |
|---|---|---|---|---|
| ***kat*G315** | *kat*G MUT1 | S315T1 | 44(91.7) | 32 (97) |
| | *kat*G MUT2 | S315T2 | 0 | 0 |
| ***inhA*** | *inhA* MUT1 | C→15T | 4 (8.3) | 0 |
| | *inhA* MUT2 | A→16G | 0 | 0 |
| | *inhA* MUT3A | T→8C | 0 | 0 |
| | *inhA* MUT3B | T→8A | 0 | 0 |
| ***kat*G315 and *inhA*** | *kat*G MUT1 and *inhA* MUT1 | S315T1 and C→15T | 0 | 1 (3) |

isolates, to PZA in seven isolates, and to EMB in five isolates. Two Hr-TB isolates were resistant to both PZA and EMB.

## Frequency and variants of INH resistance conferring mutations among Hr-TB and MDR isolates

INH resistance conferring mutations were detected in 77.4% (n = 48) of Hr-TB isolates with the LPA, of which the *kat*G315 mutation accounted for 91.7% (n = 44) and *inh*A promoter region mutation accounted for 8.3% (n = 4) of all mutations detected. All *kat*G315 mutant strains showed hybridization to the *kat*G MUT1 probe, indicating AGC→ACC substitution. Hybridization to *kat*G MUT2 probe was not observed in any *kat*G315 mutant strain corresponding to the absence of AGC→ACA mutation among Hr-TB isolates. The mutation probe for *inh*A MUT1 was detected in four isolates suggesting C→15T substitution in the *inh*A promoter region. No Hr-TB isolate showed hybridization to *inh*A MUT2, *inh*A MUT3A, and *inh*A MUT3B probes (Table 5). One Hr-TB isolate that was RIF susceptible with phenotypic DST was classified as RIF resistant by LPA as identified by the absence of the *rpo*B WT7 probe.

Among 35 MDR-TB isolates identified by phenotypic DST, resistance to INH was detected in 94.3% (n = 33) isolates by LPA. *KatG315* mutations accounted for all (100%) of mutations detected among MDR isolates. In one isolate with *kat*G315 mutation, a concomitant mutation was observed in the *inh*A promoter region at nucleic acid position -15 representing a C→15T substitution. All *kat*G315 mutant strains showed hybridization to the *kat*G MUT1 probe, indicating AGC→ACC substitution. In the remaining two MDR-TB isolates, mutations in *kat*G15 and *inh*A promoter region were not detected using LPA (Table 5).

## Discussion

The detection of INH resistance is important in the control of drug resistant TB in high TB incidence settings. In this study we measured the diagnostic performance of the GenoType MTBDR*plus* VER-2.0 in detecting INH resistant TB among isolates collected for a DRS in Ethiopia. We found the assay had significantly higher sensitivity for detecting INH in MDR-TB (94.3%) versus Hr-TB cases (77.4%) ($P$ = 0.04). Similarly, we have demonstrated a high sensitivity (96.4%) for detecting INH resistance MDR-TB cases in our previous study [18]. Other studies also showed increased sensitivity of LPA (≥90%) for the detection of INH resistance among MDR-TB cases [7, 19, 20]. In a multi-center study, the test has been found to have a sensitivity of 89.4% and a specificity of 98.9% for detecting INH resistance in cultured isolates of *M. tuberculosis* [17]. However, the analysis was conducted without separating between Hr-TB and MDR-TB strains. This indicates that the current LPA test have less than

optimal sensitivity for detecting INH resistance among Hr-TB cases delaying proper treatment and increasing risk of treatment failure.

While most INH resistance is due to mutations in the *inhA* and *katG* genes, mutations in other genetic regions such as *ahpC*, *fabG1* and *ndh* genes have also been associated with resistance. Importantly these genes are not detectable by the LPA [21]. Sequencing of resistance genes adjacent to *katG* S315T or *inhA* promoter mutations have been suggested to improve the sensitivity of LPA [22]. The lower sensitivity of LPA for detection of INH resistance among Hr-TB in this study suggests that INH resistance conferring mutations other than the canonical *katG*315 and *inhA* promoter region tend to occur more commonly among Hr-TB cases than MDR-TB cases. In line with this, the WHO recommends the use of phenotypic DST as a follow on test when LPA does not detect INH resistance, particularly in populations with a high pre-test probability of resistance to INH [7]. Thus, using GenoType MTBDR*plus* VER-2.0 LPA as stand-alone test for the detecting INH resistance among Hr-TB cases may result in the missed detection of Hr-TB cases, placing them on an ineffective treatment regimen [3].

In this study, the specificity of GenoType MTBDR*plus* VER-2.0 LPA was the same as phenotypic DST. Our previous study demonstrated that this assay had a specificity of 100% for detecting INH resistance among smear positive MDR-TB cases [18]. Similar findings have been reported across studies [7, 19].

We found *katG*315 and *inhA* mutation probes in all INH resistant phenotypes detected by LPA. Studies have shown that false INH resistance is unlikely to occur when mutation probes are detected [19, 23]. However, previously we have detected INH resistance due to absence of wildtype probes in *katG*315 locus [18]. Despite being rare, false resistance may occur when the wildtype probe disappears and the corresponding mutation probe is not detected [23]. In this circumstance, INH resistance should be inferred as the test could non-selectively detect synonymous mutation for predicting INH resistance [15]. Our finding clearly shows that detection of INH resistant mutation probes indicates true-resistant results so that patients can be prescribed appropriate treatment.

In this study, the *katG* 315 mutation was the most common mutation among Hr-TB and MDR-TB strains, yet it occurred less frequently among Hr-TB compared to MDR-TB (71% versus 94.3%). Similarly, other studies have observed differences in relative frequency of these genes between Hr-TB and MDR-TB isolates. In these studies, frequency of the *katG* mutation has been found to be lower in Hr-TB strains compared to MDR-TB strains [12, 13]. Analysis of global MTBC isolates also showed that *katG*315 mutation is the most frequent mutation observed among phenotypic INH resistant cases [11]. The dominance of *KatG* mutants could be attributed to its ability for clustering and acquiring additional compensatory mutations enhancing its transmission without negative selection pressure [24, 25]. In addition, it can be explained in part by the association of *KatG* mutation with Euro-American lineage, which is the most common lineage in the World including Ethiopia [25–27].

In this study, LPA detected one RIF resistant mutant that was classified as RIF susceptible using phenotypic DST. It has been demonstrated that some *rpoB* mutations, which are associated with low-level RIF resistance, are undetectable by a phenotypic DST at critical concentration of 1.0 μg/ml [28]. In line with this, WHO has recently lowered the critical concentration of RIF for phenotypic DST to 0.5 μg/ml to resolve discordance between genotypic and phenotypic DST [29].

Our study showed high rate of resistance to PZA and EMB among Hr-TB isolates. These drugs have been incorporated additionally in the shorter MDR-TB regimen throughout the intensive and continuation phases of the treatment duration [30]. Therefore, phenotypic DST for PZA and EMB appears to be important before treatment of Hr-TB cases. This finding also highlights the need for rapid molecular tests for these drugs.

Our study was subject to limitations. Sequencing was not included as a composite reference standard with phenotypic DST. Previous studies highlighted the contribution of sequencing for detecting INH resistance-conferring mutations that are not detectable using LPA [13, 20, 23]. In this situation, sequencing is important to shed light on INH mutants not detectable by LPA. Sequencing can also be used to avoid misclassification of true resistant as susceptible strains [17, 20]. LPA detected resistance, due to *rpo*B mutation, in RIF susceptible phenotypes has been confirmed using sequencing [28]. In this situation, the sensitivity of LPA would increase if sequencing is included as a composite reference standard. Since, detection of INH resistance mutation among INH susceptible phonotypes using LPA is very rare across studies, the use of sequencing as a composite reference for evaluating the specificity of LPA for detecting INH resistance is minimal.

## Conclusions

The sensitivity of GenoType MTBDR*plus* VER 2.0 LPA for the detection of INH resistance was lower among Hr-TB isolates compared to MDR-TB isolates in Ethiopia. The *kat*G315 mutation is the most common INH resistance conferring gene among Hr-TB and MDR-TB isolates. We recommend that additional genes be evaluated for INH resistance to improve the sensitivity of the GenoType MTBDR*plus* test in Hr-TB cases.

## Supporting information

**S1 File.**
(XLSX)

## Acknowledgments

The authors would like to thank the DRS study investigators and study participants for their contribution to this study.

## Author Contributions

**Conceptualization:** Shewki Moga.

**Data curation:** Shewki Moga, Dinka Fikadu, Getu Diriba, Misikir Amare.

**Formal analysis:** Shewki Moga, Dinka Fikadu.

**Funding acquisition:** Russell R. Kempker, Henry M. Blumberg.

**Investigation:** Shewki Moga, Dinka Fikadu, Betselot Zerihun, Getu Diriba, Russell R. Kempker, Henry M. Blumberg, Tamrat Abebe.

**Methodology:** Shewki Moga, Kidist Bobosha, Dinka Fikadu, Betselot Zerihun, Getu Diriba.

**Project administration:** Shewki Moga.

**Supervision:** Shewki Moga, Kidist Bobosha, Misikir Amare, Russell R. Kempker, Henry M. Blumberg, Tamrat Abebe.

**Validation:** Shewki Moga, Russell R. Kempker, Henry M. Blumberg, Tamrat Abebe.

**Writing – original draft:** Shewki Moga.

**Writing – review & editing:** Shewki Moga, Kidist Bobosha, Russell R. Kempker, Henry M. Blumberg, Tamrat Abebe.

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
