## [Decision Letter · Decision Letter 0]

15 Nov 2022

PONE-D-22-26863Diagnostic performance of the GenoType MTBDRplus VER 2.0 line probe assay for the detection of isoniazid resistant Mycobacterium tuberculosis in EthiopiaPLOS ONE

Dear Dr. Shewki Moga Siraj,

Thank you for submitting your manuscript to PLOS ONE. After careful consideration, we feel that it has merit but does not fully meet PLOS ONE’s publication criteria as it currently stands. Therefore, we invite you to submit a revised version of the manuscript that addresses the points raised during the review process.

We look forward to receiving your revised manuscript.

Kind regards,

Guocan Yu

Academic Editor

PLOS ONE

Journal Requirements:

Reviewers' comments:

Reviewer's Responses to Questions

**Comments to the Author**

1. Is the manuscript technically sound, and do the data support the conclusions?

Reviewer #1: Yes

Reviewer #2: Yes

Reviewer #3: Partly

2. Has the statistical analysis been performed appropriately and rigorously? 

Reviewer #1: Yes

Reviewer #2: Yes

Reviewer #3: No

3. Have the authors made all data underlying the findings in their manuscript fully available?

Reviewer #1: Yes

Reviewer #2: Yes

Reviewer #3: Yes

4. Is the manuscript presented in an intelligible fashion and written in standard English?

Reviewer #1: Yes

Reviewer #2: Yes

Reviewer #3: Yes

5. Review Comments to the Author

Reviewer #1: This study followed scientific methods to analyze observations and to present results successfully, study design is more likely to be a cross-sectional study, a design fits for studying test's performance.

Reviewer #2: This manuscript assesses the Diagnostic performance of the GenoType MTBDRplus VER 2.0-line probe assay for the detection of isoniazid resistant Mycobacterium tuberculosis in Ethiopia.

The subject remains a point of interest, however looking into the current form the manuscript seems to have major needed correction, mostly clarifying and updating the mix-ups between the methods and result section. The discussion part is well written, and a few minor edits are required there too.

Reviewer #3: Dear Editorial manager

Thank you for giving chance to review this manuscript. It is important, especially to observe the detection level of INH resistance by LPA among phenotypically Hr-TB and MDR-TB isolate. However, it needs major revision. Here below are my comment for Authors.

1. In line 40 ---the katG 315 mutation was observed in 94.3% (n=33) of MDR-TB by phenotypes, but in the Table 5 it is written, 97% similar for katG 315 mutation was observed in 71% (n=44) of Hr-

40 TB phenotypes, but not in the table.

2. Line 41-and 42 – need revision. The percent is not correct

3. Why Table 1and 2 sensitivity and NPV differed while the number of cases were similar. What is there difference also? This make also confusion to me to get the message in the result and discussion part.

4. Table 4 calculation is not right please revised it.

5. Table 5 cite reference for Nucleotide change

6. In the figure 1 among 62 INH resistance 14 were not INH resistance, but in the table 1and 2- written as 12 was not INH resistance by the LPA

7. line 232 and 233 need reference

8. What about patient characteristics?

9. While 36 phenotypically INH resistant isolates were required for assessing the sensitivity of GenoType MTBDRplus LPA VER 2.0. The Author used 62 INH resistant isolates which is good but based on what proportion the Author compere with Pan-susceptible. It is also better to calculate sample size for phenotypically MDR-TB and INH resistant isolates separately.

10. line 270-275 what is it’s important to discuss here about RIF resistant mutant

11. The study doesn’t show the diagnostic accuracy of LPA. Sensitivity, Specificity, PPV and NPV are different from accuracy

12. Why the Author left to interpret the presence and absence of wild-type probes using the LPA card.

13. It is not clear Why the Author recommend additional genes to be evaluated for INH resistance? Is that because of the detection of INH resistance was lower among Hr-TB cases compared to MDR-TB isolates. If so first it will better to calculate sample size for phenotypically INH resistant proportional and MDR-TB separately then compare each other.

14. Define INH, HPV and NPV under each table

15. No acknowledgment

16. add Author contribution

6. PLOS authors have the option to publish the peer review history of their article (what does this mean?). If published, this will include your full peer review and any attached files.

Reviewer #1: **Yes: **Layth Al-Salihi

Reviewer #2: No

Reviewer #3: **Yes: **Yeshiwork Abebaw Asaye

---

## [Author Response · Author response to Decision Letter 0]

4 Jan 2023

Reviewer #1 I have incorporated all of your suggestions into my revision. Thank you for the detailed and useful comments.

Reviewer #2 I have incorporated all of your suggestions into my revision. Thank you for the detailed and useful comments.

Dear: Reviewer #3

We have addressed your comments across the manuscript. Most of numerical related comments were very useful, so we have now applied the proposed corrections in our manuscript. Given that the paper is aimed on evaluation of diagnostic assay, we decided not to include demographic data of patients. We would like to thank you again for sparing the time to write so many detailed and useful comments. I look forward to hearing from you

---

## [Editor Report · Decision Letter 1]

19 Jan 2023

PONE-D-22-26863R1Diagnostic performance of the GenoType MTBDRplus VER 2.0 line probe assay for the detection of isoniazid resistant Mycobacterium tuberculosis in EthiopiaPLOS ONE

Dear Dr. Shewki Moga Siraj,

Thank you for submitting your manuscript to PLOS ONE. After careful consideration, we feel that it has merit but does not fully meet PLOS ONE’s publication criteria as it currently stands. Therefore, we invite you to submit a revised version of the manuscript that addresses the points raised during the review process.

We look forward to receiving your revised manuscript.

Kind regards,

Guocan Yu

Academic Editor

PLOS ONE

Additional Editor Comments:

Dear authors,

I find that you did not respond to reviewer 3's comments, please clarify. Please indicate the specific line number after the modification. Removal of modification marks, modifications can be marked in red. Please revise and resubmit, including all reviewers' comments.
---

## [Author Response · Author response to Decision Letter 1]

1 Feb 2023

Reviewer 1: I have incorporated all of your suggestions into revised manuscript. They were very helpful. Thank you

Reviewer 2: I have incorporated all of your suggestions into revised manuscript. They were very helpful. Thank you

Reviewer 3: I have addressed most points raised into my revised manuscript and put explanation for recommendation which do not require revision. They were very helpful. Thank you

---

## [Decision Letter · Decision Letter 2]

6 Mar 2023

PONE-D-22-26863R2Diagnostic performance of the GenoType MTBDRplus VER 2.0 line probe assay for the detection of isoniazid resistant Mycobacterium tuberculosis in EthiopiaPLOS ONE

Dear Dr. %Shewki Moga Siraj,

Thank you for submitting your manuscript to PLOS ONE. After careful consideration, we feel that it has merit but does not fully meet PLOS ONE’s publication criteria as it currently stands. Therefore, we invite you to submit a revised version of the manuscript that addresses the points raised during the review process.

We look forward to receiving your revised manuscript.

Kind regards,

Guocan Yu

Academic Editor

PLOS ONE

Reviewers' comments:

Reviewer's Responses to Questions

**Comments to the Author**

1. If the authors have adequately addressed your comments raised in a previous round of review and you feel that this manuscript is now acceptable for publication, you may indicate that here to bypass the “Comments to the Author” section, enter your conflict of interest statement in the “Confidential to Editor” section, and submit your "Accept" recommendation.

Reviewer #2: All comments have been addressed

Reviewer #4: (No Response)

Reviewer #5: All comments have been addressed

2. Is the manuscript technically sound, and do the data support the conclusions?

Reviewer #2: Yes

Reviewer #4: No

Reviewer #5: Yes

3. Has the statistical analysis been performed appropriately and rigorously? 

Reviewer #2: Yes

Reviewer #4: No

Reviewer #5: Yes

4. Have the authors made all data underlying the findings in their manuscript fully available?

Reviewer #2: No

Reviewer #4: Yes

Reviewer #5: Yes

5. Is the manuscript presented in an intelligible fashion and written in standard English?

Reviewer #2: Yes

Reviewer #4: Yes

Reviewer #5: Yes

6. Review Comments to the Author

Reviewer #2: This manuscript assesses the Diagnostic performance of the GenoType MTBDRplus VER 2.0-line probe assay for the detection of isoniazid resistant Mycobacterium tuberculosis in Ethiopia.

All the major and minor comments have been adequately addressed, and I have no further comments. I recommend publishing this article.

Reviewer #4: Summary

This was a retrospective study that aimed at assessing the performace of GenoType MTBDRplus version 2.0 (Hain ver. 2) in detecting isoniazid resistance among M tuberculosis complex (MTBC) clinical isolates. Samples were taken from the third Ethiopian national drug resistance survey conducted between August 2017 and December 2019. Phenotypic drug susceptibility testing (DST) with MGIT determined bacillary susceptibility to isoniazid. Phenotypic DST results were available for 1552 MTBC isolates out of 2268 isolates. Among these 1552 isolates, 62 were isoniazid-resistant and rifampicin-susceptible, and 35 were multidrug-resistant (MDR, resistant to both isoniazid and rifampicin). These 97 isoniazid-resistant isolates together with 40 randomly selected isoniazid-susceptible isolates were included in the analysis. They found that Hain ver. 2 was 77.4% sensitive and 100% specific for detecting isoniazid resistance among rifampicin-susceptible isolates, and 94.3% sensitive and 100% specific for detecting isoniazid resistance among MDR isolates. Additionally, they also calculated positive and negative predictive values (PPV and NPV) using the included samples. They concluded that Hain ver. 2 was significantly more sensitive for detecting isoniazid resistance among rifampicin-susceptible isolates than MDR isolates. They also suggested that, using Hain ver.2 as stand-alone test for detecting isoniazid resistance among rifampicin-susceptible cases may miss isoniazid-resistant cases, who would then be given an ineffective treatment regimen. They recommended that additional genes be evaluated for isoniazid resistance to improve the sensitivity of Hain ver.2 for rifampicin-susceptible TB cases.

Major comments

1. This study did not provide any information about the respective prevalence of isoniazid resistance among rifampicin-susceptible isolates and rifampicin-resistant isolates.

2. It is uncertain whether the 40 randomly selected isoniazid-susceptible isolates were rifampicin-susceptible. It may be better to evaluate the specificity of Hain ver. 2 for isoniazid resistance using isoniazid-susceptible isolates with matched bacillary susceptibility to rifampicin among rifampicin-susceptible and rifampicin-resistant isolates, respectively.

3. PPV and NPV were erroneously estimated in this study without due consideration of the pre-test probability of isoniazid resistance. The study sample of 62 isoniazid-resistant and rifampicin-susceptible isolates plus an arbitrarily determined number of randomly selected isoniazid-susceptible isolates would not give any information about the pre-test probability of isoniazid resistance among rifampicin-susceptible isolates. Neither would the study sample of 35 MDR isolates plus the same batch of isoniazid-susceptible isolates give any information about the pre-test probability of isoniazid resistance among rifampicin-resistant isolates.

4. The performance of Hain ver.2 is heavily dependent on the pre-test probability of isoniazid resistance. This is why WHO has recommended that phenotypic DST should be done when Hain ver. 2 does not detect isoniazid resistance and the pre-test probability of isoniazid resistance is high. In general, isoniazid resistance is highly correlated with rifampicin resistance. It may be reasonable to expect a prevalence of isoniazid resistance in the range of 5-15% among rifampicin-susceptible TB cases, and about 90% among rifampicin-resistant TB cases. In a theoretical population of 1000 rifampicin-susceptible TB patients, with a pre-test probability of 15% for isoniazid resistance, if Hain ver. 2 is 77.4% sensitive and 100% specific, Hain ver. 2 would detect 116 cases and miss 34 cases. There are no false-positive cases and 850 true-negative cases. PPV is 100% and NPV is 96.2%. On the other hand, in a theoretical population of 1000 rifampicin-resistant TB patients, with a pre-test probability of 90% for isoniazid resistance, if Hain ver. 2 is 94.3% sensitive and 100% specific, Hain ver. 2 would detect 849 cases and miss 51 cases. There are no false-positive cases and 100 true-negative cases. PPV is 100% and NPV is 66.2%. Thus, with reference to sensitivity and specificity values found in this study, Hain ver. 2 has excellent PPV for isoniazid-resistant case regardless of rifampicin susceptibility, and a more reliable negative result when the patient has rifampicin-susceptible TB than when the patient has MDR-TB.

5. It has been shown that, if the TB treatment regimen is properly formulated with inclusion of ethambutol, missing initial isoniazid resistance does not affect the treatment success rate among rifampicin-susceptible TB cases treated with rifampicin, ethambutol and pyrazinamide for six months in comparison with drug susceptible TB treated with the standard six-month regimen. The situation is very different when rifampicin resistance is missed.

6. There may be a typo error regarding the P value (line 39 in abstract, line 172 in results). The P value was probably 0.04 rather than 0.4.

7. In summary, study findings did not appear to substantiate the conclusion.

Minor comments

1. The Clopper-Pearson exact method was probably used in estimating the 95% confidence interval. The modified Wald method might be more accurate.

Reviewer #5: The authors have satisfactorily addressed the previous reviewers’ comments. The revised manuscript is clearer and much improved compared to the original submission. I recommend accepting the manuscript for publication. I have added a couple of very minor stylistic comments for consideration.

Line 69: Please add “the” before “added.”

Lines 78-79: Suggest “…, can additionally be performed on smear-negative clinical specimens given its higher sensitivity than version 1.0.”

Line 116: Replace “was” with “were.”

7. PLOS authors have the option to publish the peer review history of their article (what does this mean?). If published, this will include your full peer review and any attached files.

Reviewer #2: No

Reviewer #4: No

Reviewer #5: No

---

## [Author Response · Author response to Decision Letter 2]

27 Mar 2023

Reviewer #2: We very much appreciate the time and effort for reviewing our manuscript and constructive feedback. Thank you for recommending our manuscript for publication

Reviewer #4: We very much appreciate the time and effort for reviewing our manuscript and constructive feedback. We have incorporated your suggestion into our revision and thank you for consideration of our revised manuscript

Reviewer #5: We very much appreciate the time and effort for reviewing our manuscript and constructive feedback. We have incorporated all of your suggestion into our revision. Thank you for recommending our manuscript for publication

---

## [Editor Report · Decision Letter 3]

10 Apr 2023

Diagnostic performance of the GenoType MTBDRplus VER 2.0 line probe assay for the detection of isoniazid resistant Mycobacterium tuberculosis in Ethiopia

PONE-D-22-26863R3

Dear Dr. Shewki Moga Siraj,

We’re pleased to inform you that your manuscript has been judged scientifically suitable for publication and will be formally accepted for publication once it meets all outstanding technical requirements.

Kind regards,

Guocan Yu

Academic Editor

PLOS ONE
---

## [Editor Report · Acceptance letter]

14 Apr 2023

PONE-D-22-26863R3 

Diagnostic performance of the GenoType MTBDRplus VER 2.0 line probe assay for the detection of isoniazid resistant *Mycobacterium tuberculosis* in Ethiopia. 

Dear Dr. Siraj:

I'm pleased to inform you that your manuscript has been deemed suitable for publication in PLOS ONE. Congratulations! Your manuscript is now with our production department. 

Kind regards, 

on behalf of

Dr. Guocan Yu 

Academic Editor

PLOS ONE